# Naringenin Promotes Myotube Formation and Maturation for Cultured Meat Production

**DOI:** 10.3390/foods11233755

**Published:** 2022-11-22

**Authors:** Qiyang Yan, Zhuocheng Fei, Mei Li, Jingwen Zhou, Guocheng Du, Xin Guan

**Affiliations:** 1Key Laboratory of Industrial Biotechnology, Ministry of Education, Jiangnan University, Wuxi 214122, China; 2Science Center for Future Foods, Jiangnan University, Wuxi 214122, China; 3Engineering Research Center of Ministry of Education on Food Synthetic Biotechnology, Jiangnan University, Wuxi 214122, China; 4Key Laboratory of Carbohydrate Chemistry and Biotechnology, Ministry of Education, Jiangnan University, Wuxi 214122, China

**Keywords:** cultured meat, public acceptance, flavonoids, naringenin, porcine satellite cells, differentiation medium, myogenesis, estrogen receptor β, IGF-1

## Abstract

Cultured meat is an emerging technology for manufacturing meat through cell culture rather than animal rearing. Under most existing culture systems, the content and maturity of in vitro generated myotubes are insufficient, limiting the application and public acceptance of cultured meat. Here we demonstrated that a natural compound, naringenin (NAR), promoted myogenic differentiation of porcine satellite cells (PSCs) in vitro and increased the content and maturity of generated myotubes, especially for PSCs that had undergone extensive expansion. Mechanistically, NAR upregulated the IGF-1/AKT/mTOR anabolic pathway during the myogenesis of PSCs by activating the estrogen receptor β. Moreover, PSCs were mixed with hydrogels and cultured in a mold with parallel micro-channels to manufacture cultured pork samples. More mature myosin was detected, and obvious sarcomere was observed when the differentiation medium was supplemented with NAR. Taken together, these findings suggested that NAR induced the differentiation of PSCs and generation of mature myotubes through upregulation of the IGF-1 signaling, contributing to the development of efficient and innovative cultured meat production systems.

## 1. Introduction

Cultured meat is a future food technology aiming to bridge the meat supply gap by revolutionizing meat production in a way that is efficient and environmentally friendly [1,2]. As a new type of food, it is challenging, but extremely important, to have the public accept and enjoy cultured meat [3,4,5]. From the consumer’s perspective, taking in nutrients and satisfying flavor and taste are the main purposes of eating meat [6]. Protein is the most important component of meat, and meat proteins mainly include myofibrillar proteins, sarcoplasmic proteins, and matrix proteins [7]. Among them, myofibrillar proteins not only endow myofibers with unique elongated fibrillar structures, but they also act as molecular motors for muscle contraction [8]. Furthermore, myofibrillar proteins are rich in essential amino acids in proper proportions, so they can be well absorbed and utilized by the body [9]. In addition, myofibrillar proteins play a decisive role in the water-holding, gelation, and emulsification properties of meat, endowing it with a characteristic mouthfeel [10,11,12]. Therefore, cultured meat products should contain adequate myofibrillar proteins, which are essential for them to achieve superior flavor and nutrition and to be accepted by the public.

Muscle satellite cells are muscle stem/progenitor cells with the capacity to proliferate and differentiate in vitro and have been widely applied as seed cells for cultured meat [13]. During the manufacture of cultured meat, muscle satellite cells first expand to reach sufficient numbers and then differentiate and fuse into myotubes. The differentiation process is accompanied by the generation and maturation of myosin and actin, as well as myofibril assembly [14]. As the central protein of muscle, myosin accounts for 50–60% of myofibrillar proteins and determines the structure, properties, and function of myofibrils [11]. Meanwhile, myosin heavy chain (MYH) isoforms are the most commonly used markers for the characterization of myotubes. Different MYH isoforms are expressed at specific stages of muscle development: myosin heavy chain 3 (MYH3) at the embryonic stage, myosin heavy chain 8 (MYH8) at the neonatal stage, myosin heavy chain 1 (MYH1), myosin heavy chain 2 (MYH2), and myosin heavy chain 7 (MYH7) in adult muscle [15,16]. Generally, muscle satellite cells can differentiate and form myotubes in vitro under conditions of high cell density or stimulation by serum starvation [17]. However, the differentiation efficiency is not satisfactory, resulting in a very low proportion of myotubes and myofibrils in the cultured meat samples, and most proteins are still cytoplasmic proteins and nuclear proteins [18]. Therefore, the textural and nutritional qualities of current cultured meat products are inferior to that of traditional meat [19,20].

Several physical and chemical strategies have been reported to improve myosin expression and myotube formation in vitro, including physical stimulation [19] and compound induction [21,22,23]. Physical stimulation usually requires the development of specialized equipment, often leading to high costs and difficulties in large-scale applications. From the perspective of cultured meat production, medium modification is a simple and efficient way to improve production efficiency and product quality. Some growth factors, such as IGF-1 [24] and small molecular compounds such as dehydrocorydaline [25], silibinin [26], and resveratrol [27], have been reported to be beneficial for myogenic differentiation. However, most of the reported compounds only enhance embryonic MYH3 expression but fail to promote the maturation of myotubes or lack research data on more mature MYH subtypes. Therefore, it is necessary to explore more effective and safe additives to promote differentiation and increase myosin expression, accumulation, and maturation, which is of great importance for the quality improvement of cultured meat.

This study aims to find an economical and effective additive that supports the generation of a large number of highly mature myotubes in vitro. Furthermore, the relevant molecular mechanism and application strategy will be explored. These results contribute to the development of efficient myotube production systems to promote the production of high-quality cultured meat products with better public acceptance. It also provides a theoretical and technical basis for deepening muscle development.

## 2. Materials and Methods

### 2.1. Cell Culture

C2C12 myoblasts were purchased from ATCC (Shanghai, China) and cultured in the proliferation medium (PM). Porcine satellite cells (PSCs) were isolated from large white pigs, as reported previously [28], and cultured on flasks or plates coated with Matrigel (Corning Inc., Corning, NY, USA) in PM. PM was Dulbecco modified Eagle’s medium (DMEM; Thermo Scientific, Waltham, MA, USA) containing 10% fetal bovine serum (FBS; Thermo Scientific, Waltham, MA, USA) and 1% penicillin-streptomycin (P/S; Thermo Scientific, Waltham, MA, USA). Cells were passaged every 2–3 days to maintain the confluency < 60%. PSCs at passage 4 (P4) to passage (P8) were used for experiments.

### 2.2. Myogenic Differentiation

C2C12 myoblasts were seeded on culture plates in PM to achieve a 90% confluence, then the medium was changed to the basal differentiation medium (DM), supplemented with 20 µM epicatechin (EC, Sigma-Aldrich, Burlington, MA, USA), 10 µM genistein (GEN, Sigma-Aldrich), 10 µM luteolin (LUT, Sigma-Aldrich), 50 µM quercetin (QUE, Sigma-Aldrich), or 10 µM naringenin (NAR, Sigma-Aldrich) to induce differentiation. DM was DMEM containing 2% horse serum (HS; Thermo Scientific, Waltham, MA, USA) and 1% P/S. The differentiation lasted for 3 days, and the medium was changed every day.

PSCs were seeded on Matrigel-coated plates and cultured in PM for 2–3 days. When cells reached 90% confluence, differentiation was induced in DM supplemented with NAR at the indicated concentration for 5–7 days. In some experiments, ERβ antagonist PHTPP (MedChemExpress, Shanghai, China) at 10 μM was added to DM alone or in combination with NAR. The medium was changed every day.

### 2.3. Mold Fabrication and 3D Culture

The polydimethylsiloxane (PDMS) mold was imprinted from PLA mold with parallel micro-channels that were patterned using a 3D printer (Raise3d, Shenzhen, China). The mold groove area is 1 cm^2^, and the size of the micro-channel is 500 μm. Before cell seeding, the PDMS mold was soaked in a 75% ethanol solution for 2 h, washed 3 times with PBS, sterilized under UV light for 6 h, and dried completely.

PSCs were suspended at a density of 2 × 10^7^ cells/mL in DMEM containing 5 mg/mL fibrinogen and 0.25 mg/mL Matrigel, followed by adding thrombin (Sigma-Aldrich, Burlington, MA, USA) for a total enzyme activity of 1 U. After pipetting up and down multiple times, the mixture was poured onto the PDMS mold. The total volume of the cell-reagent mixture in every mold was 100 µL. After incubating at 37 °C for at least 30 min for hydrogel polymerization, the mold with hydrogels was placed in the medium for proliferation and differentiation. The fibrinolytic inhibitor aminocaproic acid (ACA; Sigma-Aldrich, Burlington, MA, USA), at a concentration of 1 mg/mL, was added to the medium to prevent hydrogel degradation.

### 2.4. MTT Assay

PSCs were seeded in 96-well plates at a density of 5 × 10^3^ cells/well cultured in PM supplemented with or without NAR at various concentrations. After 48–72 h, cells were washed with PBS twice and incubated with the MTT solution (Solebro Science & Technology Co., Ltd., Beijing, China) at 37 °C for 4 h. Then, the supernatant was aspirated, and 100 μL of DMSO was added to dissolve the formazan. Then, the absorbance values of each well were measured at 490 nm using a microplate reader (BioTek Instruments Inc., Winooski, VT, USA).

### 2.5. Cell Proliferation Assay

PSCs were seeded in T25 flasks at a density of 4 × 10^5^ cells per flask and cultured in PM supplemented with or without NAR at various concentrations. After 72 h, the cell number of each group was counted with a Countness II cell counter (Thermo Scientific, Waltham, MA, USA).

### 2.6. Immunostaining

After fixing with 4% paraformaldehyde (in PBS) at 25 °C for 15 min, cells were permeabilized with 0.5% Triton X-100 in PBS for 15 min and blocked with 1% BSA (in PBS) for 1 h. Then, cell samples were incubated with mouse anti-myosin heavy chain 3 (MYH3) antibody (Santa Cruz Biotechnology, CA, USA) or rabbit anti-Ki67 antibody (Cell Signaling Technology, Danvers, MA, USA) at 4 °C overnight and, subsequently, with CoraLite488-conjugated goat anti-mouse IgG (H + L) or CoraLite488-conjugated goat anti-rabbit IgG (H + L) (Proteintech Group, Wuhan, China) at 25 °C for 1 h. After staining with DAPI (Sigma-Aldrich, Burlington, MA, USA) for 10 min, samples were imaged with a fluorescence microscope (Mshot, Guangzhou, China). The fusion index was calculated as the percentage of nuclei in MYH3-positive myotubes (≥2 nuclei) to total nuclei within MYH3-positive myotubes [24]. At least 3 field images were analyzed per sample.

To visualize F-actin in cells cultured in three-dimensional (3D) hydrogels, the samples were first incubated with CoraLite^®^488-Phalloidin (Proteintech Group, PF00001) at 4 °C for 2 h. Then, the samples were stained with Hoechst 33342 (Beyotime, C1022) for 10 min and observed using a fluorescence microscope (Mshot, Guangzhou, China).

### 2.7. Western Blot

Cells were collected and lysed with RIPA lysis buffer (50 mM Tris, 150 mM NaCl, 0.1% SDS, 1% sodium deoxycholate (pH 7.4), Beyotime, Shanghai, China) for 30 min on ice. Where indicated, nuclear and cytoplasmic fractions were extracted using a nuclear and cytoplasmic protein extraction kit (Beyotime, Shanghai, China), according to the manufacturer’s instructions. For extracting the protein of cells differentiated in the hydrogel, samples were ground to complete lysis using a tissue grinder (Beyotime, Shanghai, China) after adding RIPA lysis buffer.

Protein samples were quantified by BCA protein assay (Beyotime, Shanghai, China), separated by 10% SDS-PAGE, and transferred to PVDF membranes. After blocking with 5% non-fat dried milk in TBS containing 0.1% Tween 20 for 1 h at 25 °C, the membranes were incubated with primary antibodies overnight at 4 °C and, subsequently, with HRP-labeled secondary antibodies (goat anti-mouse, or goat anti-rabbit, Beyotime, Shanghai, China) for 1 h at 25 °C. The following was detailed information for primary antibodies. Rabbit anti-phospho-IGF-1R, rabbit anti-IGF-1R, rabbit anti-phospho-AKT, rabbit anti-AKT, rabbit anti-phospho-mTOR, and rabbit anti-mTOR were all obtained from Cell Signaling Technology (Danvers, MA, USA); mouse anti-ERα, mouse anti-ERβ, mouse anti-IGF-1, mouse anti-MYH3, mouse anti-MYH8, mouse anti-MYH1/2, and mouse anti-MYH7 were obtained from Santa Cruz Biotechnology (Santa Cruz, CA, USA); mouse anti-α-tubulin, mouse anti-GAPDH, and rabbit anti-Histone H3 were obtained from Proteintech Group (Wuhan, China). Blots were detected with the Tanon 4600SF scanner system (Shanghai, China) and analyzed using ImageJ (National Institutes of Health, Stapleton, NY, USA) for densitometry. Protein expression was normalized to α-Tubulin or GAPDH or Histone H3, then normalized to the control samples and presented as a relative fold change with the control sample set to a value of 1.

### 2.8. Statistical Analysis

Experiments were conducted at least three times independently. Data were visualized using Prism, Ver. 8 (GraphPad Software Inc., La Jolla, CA, USA) and presented as mean ± SD. Statistical significance was defined as *p* < 0.05, which was determined by the student’s *t*-test.

## 3. Results

### 3.1. Naringenin Showed an Excellent Effect on C2C12 Myogenesis

Flavonoids widely exist in natural plants and belong to the secondary metabolites of plants. In this work, we selected five flavonoids that have been reported to promote myoblast differentiation in vitro or muscle development in vivo, including epicatechin (EC, flavanol), genistein (GEN, isoflavone), luteolin (LUT, flavone), quercetin (QUE, flavanol), and naringenin (NAR, flavanone) [22,23,29,30,31]. The effect of five compounds on in vitro myogenesis was first verified and compared with the C2C12 myoblast line using the optimal concentration reported in the literature. The results showed that all five compounds enhanced the differentiation efficiency of C2C12 cells in vitro. As shown in Figure 1a,b, the addition of NAR resulted in the greatest increase in the proportion of MYH3^+^ cells, which was 3.18-fold (SD = 0.36, *p* < 0.01) higher than the control group, followed by LUT (2.65-fold, SD = 0.77, *p* < 0.05), QUE (2.14-fold, SD = 0.28, *p* < 0.05), GEN (1. 99-fold, SD = 0.39, *p* < 0.05), and EC (1.73-fold, SD = 0.17, *p* < 0.05). Moreover, the fusion index also increased in a similar trend to MYH3 expression, with the NAR treatment achieving the most 5.70-fold (SD = 1.20, *p* < 0.01) increase (Figure 1a,c). Therefore, NAR showed the most consistent and excellent effect on myogenic differentiation among these flavonoids, so further studies on PSCs were carried out using NAR.

### 3.2. NAR Was Not Toxic for PSCs but Did Also Not Promote PSC Proliferation

We first evaluated the effects of NAR on the biological toxicity, in vitro growth, and proliferative capacity of PSCs. Ki67 protein is a commonly used marker to evaluate the cell proliferative activity, which is expressed in the G1, S, G2, and M phases of the cell cycle, but not in the G0 phase. When PSCs were exposed to various concentrations of NAR for 12 h, the proportion of Ki67^+^ cells in each NAR treatment and control group was around 20%, showing no significant difference (Figure 2a,b). Besides, PSCs were cultured in PM supplemented with 0 to 100 μM of NAR for three days, and the absolute cell number was recorded. The results showed that after three days of culture, there was no significant difference in the absolute number of all groups, which was about 1.1 × 10^6^ (Figure 2c). Meanwhile, the cell viability was measured by the MTT assay. The MTT absorbance was similar for all groups (Figure 2d), indicating that NAR was not cytotoxic to PSCs up to 100 μM. Taken together, our data suggested that NAR did not show a proliferative effect on PSCs in vitro, but it was not cytotoxic to PSCs.

### 3.3. NAR Promoted Myogenic Differentiation of PSCs at Various Passages

Subsequently, we examined the effect of NAR on the formation and maturation of myotubes from PSCs of various passages during ex vivo expansion. MYH3 immunofluorescence staining showed that the myogenic differentiation ability was weakened with the increase in cell expansion and passaging, but treatment with NAR at 10, 20, and 50 μM all upregulated the differentiation efficiency of PSCs at each passage significantly. Among the three concentrations, 20 μM NAR showed the optimum effect, with the myogenic fusion index of P4, P6, and P8 PSCs reaching 183% (SD = 7%, *p* < 0.001), 207% (SD = 10%, *p* < 0.001), and 278% (SD = 21%, *p* < 0.001) of the control group, respectively (Figure 3a,b). Furthermore, we detected the expression of two myosin heavy chain (MYH) isoforms, MYH3 (generated at the embryonic stage), and MYH8 (generated at the neonatal stage). The results showed that the addition of 20 μM NAR during the differentiation of P4, P6, and P8 PSCs increased the expression of both MYH3 and MYH8 significantly and consistently. The expression levels of MYH3 and MYH8 proteins in the 20 μM NAR group were more than 150% of that in the control group (Figure 3c–e). These results suggested that NAR could promote the synthesis and maturation of myosin in myotubes differentiated from expanded myoblasts.

### 3.4. NAR Upregulated ERβ and IGF-1 Signaling during Myogenesis of PSCs

NAR has been known as a phytoestrogen whose chemical structure is similar to estrogen, and it can activate estrogen receptors (ERs) [32]. We first examined the influence of NAR on the expression of ER isoforms in PSCs. The results showed that all groups had no significant difference in the expression of ERα (Figure 4a,c), but NAR supplementation markedly increased the expression levels of ERβ in whole cells and the nucleus. In particular, PSCs treated with 20 μM NAR showed significantly elevated expression of ERβ in whole cells and the nucleus, which was 178% (SD = 34%, *p* < 0.05) and 173% (SD = 15%, *p* < 0.001) of that without NAR treatment, respectively (Figure 4b,d). It suggested that NAR enhanced the biosynthesis and activity of ERβ during myogenic differentiation.

IGF-1 is a well-recognized cytokine for inducing myogenic differentiation and muscle hypertrophy in vivo and cultured cells [33]. Therefore, we next examined the effect of NAR on the expression of endogenous IGF-1 and its downstream AKT/mTOR signaling in differentiating PSCs. Consistent with the expression of MYH and ERβ proteins, 20 μM of NAR showed the optimum up-regulating effect on endogenous IGF-1 and its downstream signaling pathway. Treatment of PSCs with 20 μM NAR increased the expression of endogenous IGF-1 to 175% (SD = 18%, *p* < 0.01) of the control group (Figure 4e,g) and upregulated the phosphorylation levels of IGF-1R, AKT, and mTOR proteins to 229% (SD = 69%, *p* < 0.05), 223% (SD = 67%, *p* < 0.05), and 236% (SD = 32%, *p* < 0.01) of the control group, respectively (Figure 4f,h–j). Collectively, these results demonstrated that NAR increased the level of nuclear ERβ and up-regulated the IGF-1/AKT/mTOR anabolic pathway during the myogenesis of PSCs.

### 3.5. Inhibition of ERβ Impaired the Effect of NAR on IGF-1 Signaling and Myogenesis

To further clarify the significance of ERβ in NAR-induced myogenic differentiation, we used an ERβ antagonist, PHTPP, to inhibit the activation of ERβ and then evaluated the changes in the fusion index after differentiation. The addition of PHTPP significantly reduced the levels of ERβ in the nuclei of ctrl and NAR-treated PSCs by 33% (SD = 15%, *p* < 0.01) and 61% (SD = 28%, *p* < 0.01) of the original levels, respectively (Figure 5a,b). On this basis, we analyzed the expression of IGF-1 and key molecules of IGF-1/AKT/mTOR signaling. The expression of endogenous IGF-1 in PSCs treated with the combination of PHTPP and NAR was 67% (SD = 7%, *p* < 0.01) of that in PSCs treated with NAR alone (Figure 5c,e). Similarly, the phosphorylation levels of IGF-1R, AKT, and mTOR proteins all decreased obviously to less than 70% of the original level upon the addition of PHTPP (Figure 5d,f–h). These results indicated that antagonizing ERβ impaired the effect of NAR on activating the IGF-1 signaling in PSCs during myogenesis.

Furthermore, we observed that, compared with PSCs treated with NAR alone, combined exposure of PHTPP and NAR markedly inhibited myotube formation during PSC differentiation and reduced the fusion index from 57% (SD = 3%) to 43% (SD = 1%) (Figure 5i,j), indicating that activation of ERβ was required for NAR to exert the myogenic-promoting effect. Our study showed that, during the differentiation of PSCs in vitro, NAR mediated nuclear transfer of ERβ signaling to upregulate the expression of IGF-1 and promoted the activation of IGF-1/Akt/mTOR anabolic pathway, thereby regulating myogenic differentiation and boosting myotube maturation.

### 3.6. NAR Supported the Generation of Mature Myotubes in a 3D Differentiation Condition

To determine the effect of NAR on producing PSCs-derived myotubes for cultured meat, we fabricated PDMS molds with parallel micro-channels for the manufacture of cultured pork samples (Figure 6a). PSCs were mixed with hydrogels and cultured in the mold for expansion and differentiation in a three-dimensional condition. After seven days of differentiation, with or without NAR, the generated myotubes were observed by MYH3 immunofluorescence, and the length between the long axis of myotubes was measured with the ImageJ software. The data showed that the length of generated myotubes by the induction of NAR in the mold was mainly distributed in the range of 151–200 μm, significantly longer than that of the control group, which was mostly less than 150 μm (Figure 6b,c). Furthermore, distinct sarcomere microstructures could be observed in NAR-induced myotubes but not in control myotubes (Figure 6d). Sarcomeres are repetitive sections as contractile units of myofibers, and they are representative indicators of mature myofibers [34]. Therefore, these data showed that NAR was beneficial for the manufacture of cultured meat by promoting PSCs to differentiate into myotubes and mature into myofibers.

Moreover, the protein expression of various MYH isoforms was analyzed for the assessment of myosin content and maturity. In addition to embryonic MYH3 and neonatal MYH8 isoforms, we detected MYH1, MYH2, and MYH7 isoforms, which were formed in adult muscle [15]. As expected, NAR treatment elevated the expression levels of three MYH isoforms significantly. Compared to the control group, the expression of MYH3 was 1.98-fold higher (SD = 0.29, *p* < 0.05), MYH8 was 2.01-fold higher (SD = 0.27, *p* < 0.01), and MYH1/2 was 1.45-fold higher (SD = 0.12, *p* < 0.01) in NAR-differentiated samples (Figure 6e–i). Conclusively, under the three-dimensional differentiation condition, NAR was proved to promote the differentiation and maturation of PSCs and increase the content and maturity of generated myosin, similar to the two-dimensional (2D) culture. Therefore, NAR would be an important additive for developing an effective, safe, and economical differentiation medium for the production of high-quality cultured meat.

## 4. Discussion

In recent years, the cultured meat industry has grown rapidly and made encouraging breakthroughs [17,35]. In the production process of cultured meat, the differentiation step is crucial to transform cells into edible meat. From publications in the cultured meat research field, there are relatively few studies focusing on the myogenic differentiation process and related regulation methods. Tom et al. added IGF-1 to the differentiation medium of bovine muscle stem cells and observed that the number of myotubes increased to 3.9 times [21]. Guan and colleagues found that luteolin activated the PI3K/Akt/mTOR signaling pathway to promote the expression of MYH3 [23]. Guo et al. proved quercetin had the pro-differentiation effect and upregulated the expression of MYH by 4.73-fold compared with the control group [22]. However, there is a large gap in research concerning the maturation of cultured myofibers. One of the few reports is that Furuhashi et al. engineered bovine muscle tissue containing aligned and mature myotubes by applying electrical stimulation during differentiation [19].

Previously, flavonoids have been reported to promote skeletal muscle repair and inhibit muscle atrophy in mouse models [36,37,38], implying their potential in promoting the maturation of myoblasts in vitro. Lee et al. found 20 μM of EC enhanced myogenic differentiation of C2C12 cells through stimulation of promyogenic signaling pathways, p38MAPK and Akt [29]. Gan et al. proved that GEN treatment promoted myoblast differentiation at a dose of 10 µM [30]. In addition, QUE, LUT, and NAR were also reported to be effective in promoting differentiation of different myoblast cells [22,23,31]. In this work, we first verified and compared the effect of five flavonoids on inducing myogenesis in vitro and found that NAR showed the most significant effect. NAR is a flavanone compound widely existing in tomato, grapefruit, and citrus fruits and is an approved food additive [39,40]. Our experimental results also proved that NAR was not cytotoxic to PSCs at up to 100 μM, implying the safety of its application in cultured meat production. Moreover, in the two-dimensional (2D) culture, we found 20 μM NAR promoted the myogenesis of PSCs at different passages and significantly increased the expression of MYH8, indicating the generation of mature myotubes.

ERs are widely expressed in skeletal muscle and myoblasts [41]. In general, ERs and estrogen form dimers and then bind to estrogen receptor binding elements to stimulate target gene transcription and regulate cell proliferation or differentiation [42,43]. ERs are mainly composed of two isoforms, estrogen receptor α (ERα) and estrogen receptor β (ERβ), both of which have been reported to be involved in muscle atrophy [44], muscle regeneration, and hypertrophy [45,46,47]. In this work, it was found that the pro-myogenesis role of NAR was related to ERβ. ERβ signaling is an important factor in regulating skeletal muscle and muscle satellite cells. Seko and colleagues indicated ERβ was indispensable for muscle growth and regeneration in female mice [48]. Ogawa et al. found daidzein down-regulated the expression of ubiquitin-specific protease 19 through ERβ and increased skeletal muscle mass in young female mice [49]. Velders et al. proved that ERβ regulated anabolic pathway, muscle satellite cell proliferation, and immune response to affect the growth and regeneration of skeletal muscle in mice [45].

It has been reported that agonists targeting ERβ possessed anabolic potency and stimulated the expression of endogenous IGF-1 [45]. IGF-1 is considered to be a factor closely related to skeletal muscle growth and hypertrophy. It can regulate protein synthesis and degradation, affect mitochondrial autophagy, and activate calcium signaling mediated by calcineurin [24,33,50]. By antagonizing ERβ with PHTPP, our study showed that, during the differentiation of PSCs in vitro, NAR mediated nuclear transfer of ERβ signaling to upregulate the expression of IGF-1 and promote the activation of IGF-1/Akt/mTOR anabolic pathway, thereby regulating myogenic differentiation and boosting myotube maturation.

The manufacture of cultured meat is inseparable from the three-dimensional cell culture. At present, the three-dimensional modeling methods for cultured meat include three-dimensional printing [20,51,52], plant protein scaffold [21], hydrogels [53,54], and microcarriers [55,56]. Different molding methods have their advantages and disadvantages. For example, plant protein scaffold makes the texture and appearance of the finished product more similar to that of meat, but its biocompatibility needs further study. Hydrogels have been widely applied for three-dimensional cell culture due to their excellent cell compatibility and ease of handling. In this work, we developed a three-dimensional myogenic differentiation system by mixing PSCs with hydrogels and culturing them in a PDMS mold with micro-channels, which is of benefit for the formation of aligned myotubes. Under the three-dimensional differentiation condition, NAR was proved to promote the differentiation and maturation of PSCs and to increase the content and maturity of generated myosin, similar to the two-dimensional culture. Compared with the control group, the addition of NAR significantly increased the expression of different subtypes of MYH protein, and obvious sarcomere structures were observed. Therefore, NAR would be an important additive for developing an effective, safe, and economical differentiation medium for the production of high-quality cultured meat.

## 5. Conclusions

We showed that NAR has a significant role in promoting the maturation of myotubes derived from PSCs in vitro and increasing their myosin content, and it could fill the gap in the research on the maturation of myotubes in the field of cultured meat. We certified that NAR-mediated ERβ improves the expression of IGF-1 and then activates the protein synthesis pathway, which lays a theoretical foundation for the potential application of NAR in the cultured meat industry.

## Figures and Tables

**Figure 1 foods-11-03755-f001:**
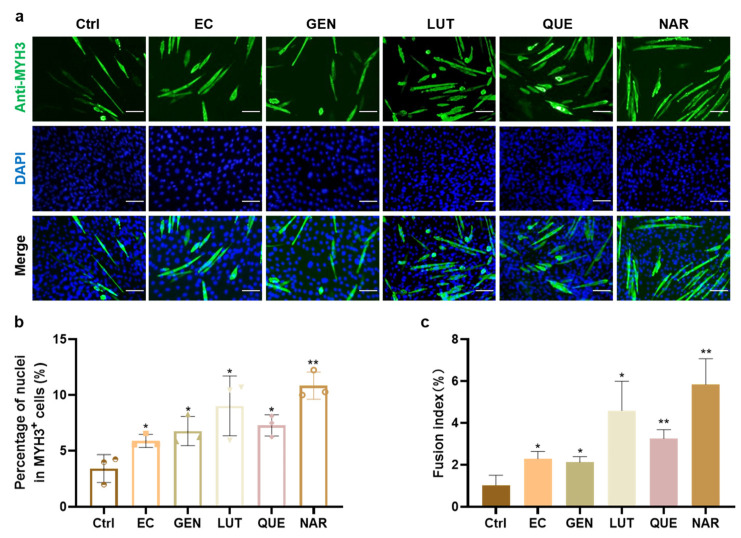
The effect of various flavonoids on myogenic differentiation of C2C12 cells. (**a**) Representative fluorescence images of C2C12 cells after three days of culture in DM alone (Ctrl), or with supplementation of EC (20 μM), GEN (10 μM), LUT (10 μM), QUE (50 nM), or NAR (10 µM). Green, anti-MYH3; Blue, DAPI, Scale bar, 100 µm. (**b**) The percentage of nuclei in myotubes (MYH3^+^ cells) to total nuclei according to immunofluorescent images from three random views. (**c**) Fusion index as a percentage according to immunofluorescent images. Error bars represent mean ± SD. The asterisk indicates a significant difference between ctrl and the treatment group. * *p* < 0.05, ** *p* < 0.01.

**Figure 2 foods-11-03755-f002:**
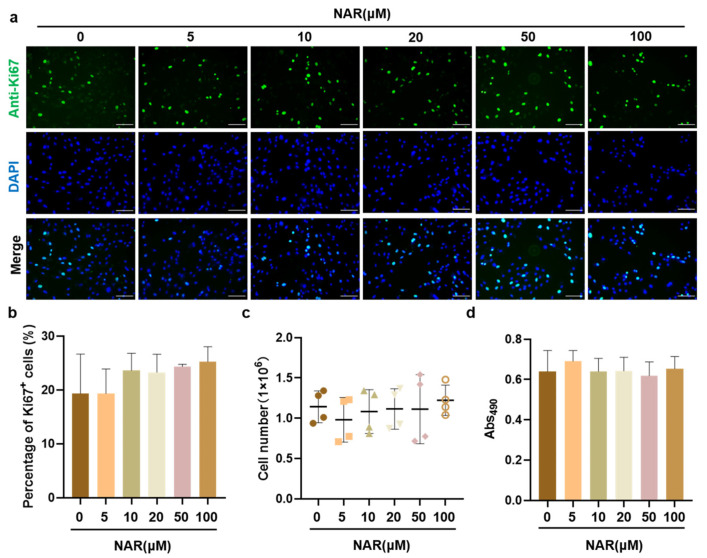
The effect of NAR on the proliferation and viability of PSCs. (**a**) Representative Ki67 fluorescence images of PSCs after 12 h of culture in PM supplemented with NAR (0, 5, 10, 20, 50, or 100 µM). Scale bar, 100 µm. (**b**) The percentage of Ki67^+^ cells calculated according to immunofluorescent images from three random views. (**c**) Absolute number of PSCs after three days of culture in PM supplemented with NAR (0, 5, 10, 20, 50, or 100 µM). (**d**) Cell viability by MTT assay. Error bars represent mean ± SD.

**Figure 3 foods-11-03755-f003:**
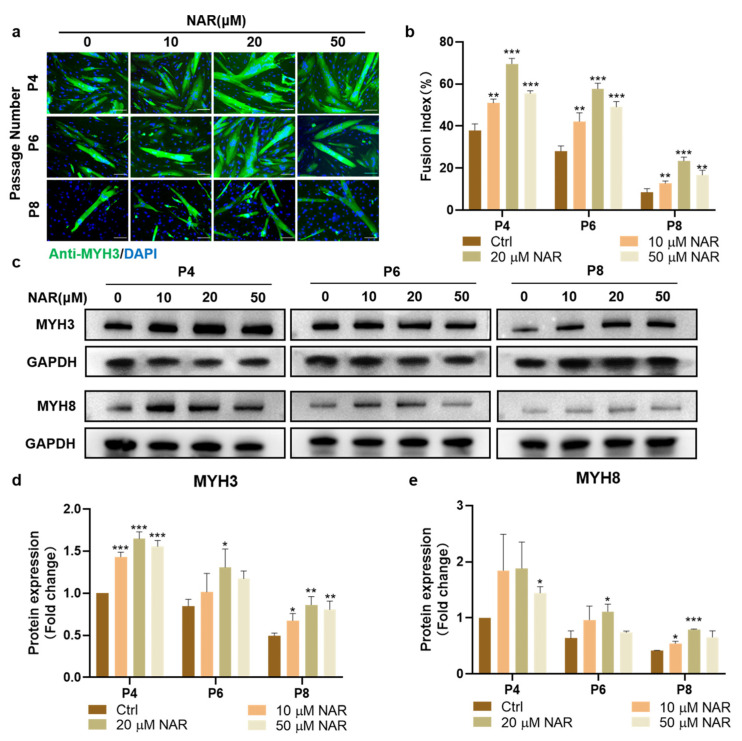
NAR increased myogenesis of PSCs at different passages. (**a**) Representative immunofluorescent images of differentiating PSCs at different passages after three days of induction in differentiation medium supplemented with NAR (0, 10, 20, 50 µM). Green, anti-MYH3; blue, DAPI. Scale bar, 100 µm. (**b**) Fusion index as a percentage according to immunofluorescent images (*n* = 3 biological replicates). (**c**) Western blot analysis of MYH expression in differentiating PSCs at different passages after three days of differentiation in the absence or presence of NAR (10, 20, 50 µM). (**d**,**e**), Quantification of MYH3 and MYH8 expression according to (**c**). The bar chart presents the quantification of blot band intensity, normalized to GAPDH levels, and presented as a fold change relative to the ctrl sample of P4, which was set to one. Error bars represent mean ± SD. The asterisk indicates a significant difference from the ctrl group at the same passage. * *p* < 0.05, ** *p* < 0.01, *** *p* < 0.001.

**Figure 4 foods-11-03755-f004:**
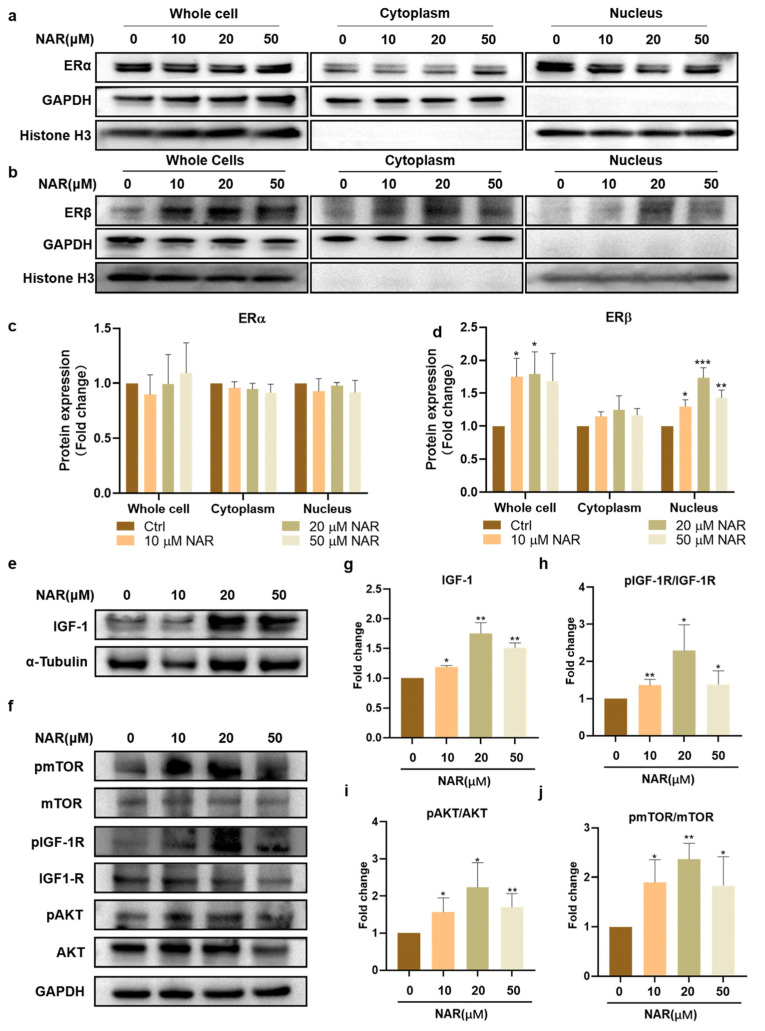
NAR upregulated ERβ and IGF-1 signaling during myogenesis. (**a**,**b**) Representative blots of ERα, ERβ, GAPDH, and Histone H3 in whole cells, cytoplasm, and nucleus. Protein was extracted from P6 PSCs after three days of differentiation in the absence or presence of NAR (10, 20, 50 µM). (**c**,**d**) Quantification of Erα and ERβ ratios according to (**a**,**b**). The bar chart presents the quantification of blot band intensity, normalized to GAPDH or histone H3 levels, and presented as fold change relative to the sample without treatment of NAR (*n* = 3 biological replicates). (**e**,**f**) Representative blots of IGF-1, IGF-1R, pIGF-1R, mTOR, pmTOR, AKT, pAKT, GAPDH, and α-tubulin in differentiating P6 PSCs after three days of induction in the absence or presence of NAR (10, 20, 50 µM). (**g**) Quantification of IGF-1 expression according to (**e**). The bar chart presents the quantification of blot band intensity, normalized to α-tubulin levels, and presented as fold change relative to the sample without treatment of NAR (*n* = 3 biological replicates). (**h**–**j**) Quantification of pmTOR/mTOR, pIGF1-R/IGF-1R, and pAKT/AKT ratios according to (**f**). The bar chart presents the quantification of blot band intensity, normalized to GAPDH levels, and presented as fold change relative to the ctrl sample (*n* = 3 biological replicates). Error bars represent mean ± SD. The asterisk indicates a significant difference from the ctrl (0 µM NAR) group. * *p* < 0.05, ** *p* < 0.01, *** *p* < 0.001.

**Figure 5 foods-11-03755-f005:**
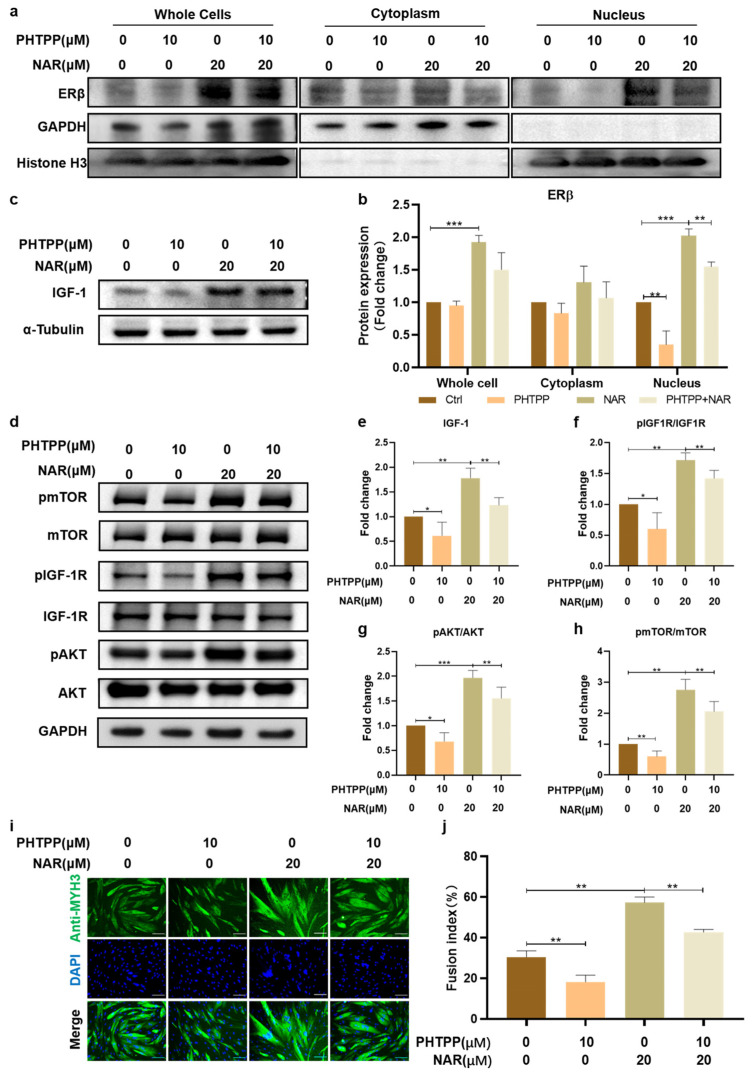
Inhibition of ERβ impaired the effect of NAR on IGF-1 signaling and myogenesis. (**a**) Representative blots of ERβ, GAPDH, and histone H3 in whole cells, cytoplasm, and nucleus. Protein was extracted from P6 PSCs after three days of differentiation with or without supplementation of NAR (20 μM) and/or PHTPP (10 μM). (**b**) Quantification of ERβ ratios according to (**a**). The bar chart presents quantification of blot band intensity, normalized to GAPDH or histone H3 levels, and presented as fold change relative to the sample without treatment of NAR or PHTPP (*n* = 3 biological replicates). (**c**,**d**) Representative blots of IGF-1, IGF-1R, pIGF-1R, mTOR, pmTOR, AKT, pAKT, GAPDH, and α-tubulin in differentiating P6 PSCs after three days of induction with or without supplementation of NAR (20 μM) and/or PHTPP (10 μM). (**e**) Quantification of IGF-1 expression according to (**c**). The bar chart presents the quantification of blot band intensity, normalized to α-tubulin levels, and presented as fold change relative to the sample without treatment of NAR or PHTPP (*n* = 3 biological replicates). (**f**–**h**) Quantification of pmTOR/mTOR, pIGF1-R/IGF-1R, and pAKT/AKT ratios according to (**d**). The bar chart presents the quantification of blot band intensity, normalized to GAPDH levels, and presented as fold change relative to the sample without treatment of NAR or PHTPP (*n* = 3 biological replicates). (**i**) Representative immunofluorescent images of differentiating P6 PSCs after three days of induction with or without supplementation of NAR (20 μM) and/or PHTPP (10 μM). Green, anti-MYH3; blue, DAPI. Scale bar, 100 µm. (**j**) Fusion index as a percentage according to immunofluorescent images (*n* = 3 biological replicates). Error bars represent mean ± SD. * *p* < 0.05, ** *p* < 0.01, *** *p* < 0.001.

**Figure 6 foods-11-03755-f006:**
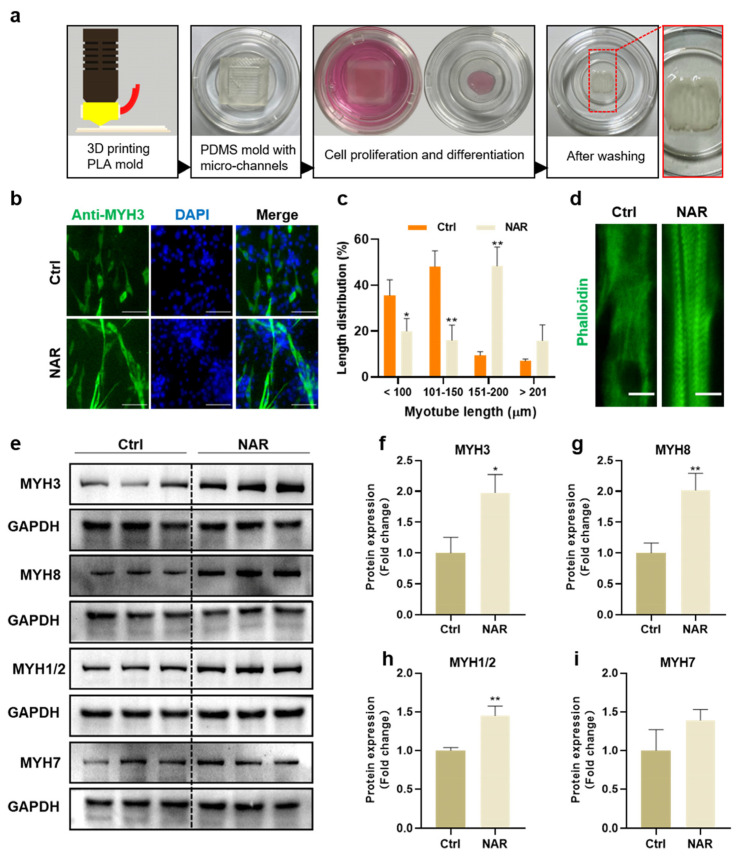
NAR promoted the generation of mature myotubes in a three-dimensional differentiation condition. (**a**) A schematic diagram of mold fabrication and cell culture for cultured pork manufacture. (**b**) Representative immunofluorescent images of differentiating myotubes after seven days of culture in differentiation medium with or without NAR (20 μM). Green, anti-MYH3; blue, DAPI. Scale bar, 100 µm. (**c**) Myotube length distribution as a percentage according to immunofluorescent images (*n* = 3 biological replicates). (**d**) Representative immunofluorescent images of sarcomere microstructures. Green, phalloidin; blue, DAPI. Scale bar, 30 µm. (**e**) Representative blots of MYH3, MYH8, MYH7, MYH1/2, and GAPDH proteins from cultured pork samples, which were differentiated with or without NAR (20 μM) for seven days. (**f**–**i**) Quantification of MYH3, MYH8, MYH7, and MYH1/2 expression normalized to GAPDH from Western blot gels and presented as fold change relative to the ctrl sample. Error bars represent mean ± SD (*n* = 3 biological replicates). The asterisk indicates a significant difference from the Ctrl group. * *p* < 0.05, ** *p* < 0.01.

## Data Availability

The data presented in this study are available upon request from the corresponding author.

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
