# Peer review of "Naringenin Promotes Myotube Formation and Maturation for Cultured Meat Production"

_foods, 2022, doi:10.3390/foods11233755_

Round 1

Reviewer 1 Report

I strongly support research aimed at reducing the production of slaughter animals and the development of high nutritional value protein technology. In fact, one of the most important sensory characteristics is texture, hence it is necessary to solve the problem of formation of myofibril structures of cultured meat. Therefore, it is necessary to explore more effective additives that promote differentiation and maturation of myosin. Searching for factors stimulating proliferation and differentiation, even in raw materials of plant origin, is necessary, although time- and labor-intensiv. The work is prepared very diligently, and its purpose is well thought out.

Author Response

Dear Reviewer,

We would like to thank you for your time and efforts in reviewing our manuscript. We also appreciate your affirmation of our work. The development of production technology of high-quality substitute protein is of great significance. We hope that this manuscript aimed at improving the quality of cultured meat can be published in Foods journal and shared with a wider audience.

Reviewer 2 Report

Foods-2005162

This manuscript investigates the effect of adding Naringenin to cultured muscle cells in the differentiation phase resulting in an increased expression of maturation genes of the myocine heavy chain isoforms as well as fusion index. A likely mechanism through the IGF-1/AKT/mTOR pathway is also suggested/demonstrated. This is a well-constructed manuscript with sound hypothesis, experimental setup and data presentation followed by relevant conclusions. Minor general and specific points is suggested below.

General points:

Assays are described as carried out on PSCs with /without Naringenin but some of the results are obtained in C2C12 cells and with several flavonoids. This should be addressed in the materials and methods section.

In all figures are significant difference indicated by horizontal bars between the different treatments and an indication of the level of significance by number of stars. A more simple way would be to indicate the stars above the bar and write in the legend that stars indicate difference from the control/control within passage. This would not work for figure 5 though

Specific points:

Line 14: I would suggest to change the word “breeding” with “rearing” or “production” as breeding is often used when referring genetic selection and “just” production.

Line 21: suggest deleting “able to be”

Figure 1: why is different concentrations of the flavonoids used? It is very likely a concentration optimum (bell curve as also indicated in figure 3). Hence, it cannot be ruled out that e.g. the lower concentration of quercetin (50 nM) makes it less effective than Nargining at 10 uM. Has concentration effects been tested prior to these with a chosen optimum for each?

Line 208: The term “safety profile” is not appropriate here. I will suggest “Nar was not toxic for PSCs but did also not promote PSC proliferation”

Figure 2 legend: suggest to swap “with” and “without” as the supplementation follows…

Line 262: change “3a, c” to “4a, c”

Line 263: change “3b, d” to “4b, d”

Line 297: change “obviously” to “significantly”

Line 346: change “obvious” to “significant”

Figure 6b is very blurred – would it be better with bigger photos?

Line 408: add “structures” after “sarcomere” to become “sarcomere structures”

Author Response

Dear Reviewer,

We would like to thank you for your time and efforts in reviewing our manuscript. We have carefully considered all your comments and have revised the manuscript according to your suggestions. The quality of the revised manuscript has been greatly improved. We hope that the revised manuscript can be accepted for publication in Foods. The point-by-point responses to your comments are as follows:

1) Assays are described as carried out on PSCs with /without Naringenin but some of the results are obtained in C2C12 cells and with several flavonoids. This should be addressed in the materials and methods section.

Response:

Thanks for your suggestions. We are sorry for missing the description of operating C2C12 cells in the materials and methods section. We have added a relevant description in the revised manuscript “C2C12 myoblasts were seeded on culture plates in PM to achieve a 90% confluence, then the medium was changed to the basal differentiation medium (DM) supplemented with 20 µM epicatechin (EC, Sigma-Aldrich, Burlington, MA, USA), 10 µM genistein (GEN, Sigma-Aldrich), 10 µM luteolin (LUT, Sigma-Aldrich), 50 µM quercetin (QUE, Sigma-Aldrich), or 10 µM naringenin (NAR, Sigma-Aldrich) to induce differentiation. DM was DMEM containing 2% horse serum (HS; Thermo Scientific, Waltham, MA, USA) and 1% P/S. The differentiation lasted for 3 days and the medium was changed every day.” (Lines 93-99)

2) In all figures are significant difference indicated by horizontal bars between the different treatments and an indication of the level of significance by number of stars. A simpler way would be to indicate the stars above the bar and write in the legend that stars indicate difference from the control/control within passage. This would not work for figure 5 though.

Response:

Thank you for your kind advice. According to your suggestion, we have deleted the horizontal bars between different groups and only used asterisk(s) to indicate significance levels in the revised Figures 1-4 and 6. Moreover, the meanings of asterisks were indicated in the corresponding figure legends.

3)

  • Line 14: I would suggest to change the word “breeding” with “rearing” or “production” as breeding is often used when referring genetic selection and “just” production.
  • Line 21: suggest deleting “able to be”
  • Line 208: The term “safety profile” is not appropriate here. I will suggest “Nar was not toxic for PSCs but did also not promote PSC proliferation”
  • Figure 2 legend: suggest to swap “with” and “without” as the supplementation follows…
  • Line 262: change “3a, c” to “4a, c”
  • Line 263: change “3b, d” to “4b, d”
  • Line 297: change “obviously” to “significantly”
  • Line 346: change “obvious” to “significant”
  • Line 408: add “structures” after “sarcomere” to become “sarcomere structures”.

Response:

Thank you for your kind suggestion. We are very sorry for these errors. We have corrected these spelling errors and mistakes in the revised manuscript.  

4) Figure 1: why is different concentrations of the flavonoids used? It is very likely a concentration optimum (bell curve as also indicated in figure 3). Hence, it cannot be ruled out that e.g. the lower concentration of quercetin (50 nM) makes it less effective than Naringenin at 10 uM. Has concentration effects been tested prior to these with a chosen optimum for each?

Response:

Thank you for your comments. It is a very good point. Before the experiments, we searched a large amount of literature and summarized five flavonoids that have been reported to promote myoblast differentiation in vitro or muscle development in vivo. Therefore, we initially intended to verify and compare the effects of these flavonoids on myogenic differentiation using the optimal concentration reported for each flavonoid. Corresponding research articles on the five flavonoids are summarized as follows:

  • Lee et al. showed that epicatechin at an optimum concentration of 20 µM enhanced myogenic differentiation of C2C12 cells through stimulation of promyogenic signaling pathways p38MAPK and Akt.

(Lee, S.J.; Leem, Y.E.; Go, G.Y.; Choi, Y.; Song, Y.J.; Kim, I.; Kim, D.Y.; Kim, Y.K.; Seo, D.W.; Kang, J.S.; et al. Epicatechin elicits MyoD-dependent myoblast differentiation and myogenic conversion of fibroblasts. PLoS One 2017, 12, e0175271.)

  • Gan et al. proved that genistein treatment at the best dose of 10 µM promoted myoblast differentiation.

(Gan, M.; Yang, D.; Fan, Y.; Du, J.; Shen, L.; Li, Q.; Jiang, Y.; Tang, G.; Li, M.; Wang, J.; et al. Bidirectional regulation of genistein on the proliferation and differentiation of C2C12 myoblasts. Xenobiotica 2020, 50, 1352-1358.).

  • Guan et al. reported that luteolin at the optimum concentration of 10 µM enhanced the migration and differentiation of myoblasts, as indicated by improved migration rate and fusion index, as well as upregulated expression of Myogenin and MyHC.

(Guan, X.; Pan, Z.; Xu, Z.; Zhang, S.; Tang, H.; Du, G.; Zhou, J. Natural flavonoid luteolin promotes the differentiation of porcine myoblasts through activation of PI3K/Akt/mTOR signaling. Food Bioscience 2022, 47, 101766.)

  • Guo et al. found that quercetin had a significant impact on myoblast differentiation with an optimum concentration of 50 nM.

(Guo, Y.; Ding, S.-J.; Ding, X.; Liu, Z.; Wang, J.-L.; Chen, Y.; Liu, P.-P.; Li, H.-X.; Zhou, G.-H.; Tang, C.-B. Effects of selected flavonoids on cell proliferation and differentiation of porcine muscle stem cells for cultured meat production. Food Research International 2022, 160, 111459.)

  • Pellegrini et al. found that naringenin promotes the myogenic differentiation of L6 myoblasts through antioxidation activity, and its optimal concentration was 10 µM.

(Pellegrini, M.; Bulzomi, P.; Galluzzo, P.; Lecis, M.; Leone, S.; Pallottini, V.; Marino, M. Naringenin modulates skeletal muscle differentiation via estrogen receptor α and β signal pathway regulation. Genes Nutr 2014, 9, 425.)

We have added the relevant explanation in the revised manuscript “In this work, we selected five flavonoids that have been reported to promote myoblast differentiation in vitro or muscle development in vivo, including epicatechin (EC, flavanol), genistein (GEN, isoflavone), luteolin (LUT, flavone), quercetin (QUE, flavanol), and naringenin (NAR, flavanone) [22,23,29-31]. The effect of five compounds on in vitro myogenesis was first verified and compared with the C2C12 myoblast line using the optimal concentration reported in the literature.” (Lines182-187)

5) Figure 6b is very blurred – would it be better with bigger photos?

Response:

Thank you for your kind suggestion. We have improved the resolution and enlarged the representative local area of Figure 6b to exhibit them more clearly in the revised manuscript.

Reviewer 3 Report

Report on the manuscript foods-2005162 entitled: Naringenin promotes myotube formation and maturation for cultured meat production

-          The authors chose to separate Results and Discussion into two different sections. Therefore, cites from the Results must be removed, and moved to the Discussion section, which must be greatly (quantitatively) improved.

-          Figures are extremely accurate and precise. But, is that really necessary? Especially when the information of the figures is not properly described and discussed later…

-          Please, consider the use of the same structure for abbreviations. Why Nar and not NAR? The authors use PSC, MYH, etc… but not Psc, Myh…

-          Lines 76-78. This paragraph does not belong to the Introduction. In this paragraph, the objective or the aim of the study must be described.

-          Line 24. Porcine myotubes? How do the authors know that the myotubes are “porcine”?

-          Lines 110, 117, 120, etc… Please, add a space between the number and the units.

-          Lines 89-176. The authors use the phrase “room temperature” too lightly.

-          Lines 184-198. Further description of Figure 1 results and critical discussion must be included. Or have the authors published the comparison among flavonoids elsewhere?

-          Lines 209-220. Further description of Figure 2 results and critical discussion must be included.

-          Lines 222-227. Treatment “0” is missing.

-          Lines 229-323. Further description of Figures 3-5 results and critical discussion must be included.

-          Figure 3. d) and e) SD bar is missing in Control.
Same Figures 4 and 5.

-          Cite format of the Reference list must be reviewed. It does not follow the journal requirements.

Author Response

Dear Reviewer,

We would like to thank you for your time and efforts in reviewing our manuscript. We have carefully considered all your comments and have revised the manuscript according to suggestions. Besides, the manuscript has been checked by a native English-speaking colleague for extensive English revisions. The quality of the revised manuscript has been greatly improved. We hope that the revised manuscript can be accepted for publication in Foods. The point-by-point responses to your comments are as follows:

1) The authors chose to separate Results and Discussion into two different sections. Therefore, cites from the Results must be removed, and moved to the Discussion section, which must be greatly (quantitatively) improved.

Response:

Thanks for your comments. According to your kind suggestions, we have deleted most of the cites in the Results section and moved explanatory sentences and related citations from the Results section to the Discussion section. Only a few indispensable references were retained to explain the detection index so that the readers can understand better. Moreover, we have added further descriptions of the data and critical discussions in the revised manuscript. The manuscript has been improved in quantity and quality, and we hope to receive your approval.

2) Figures are extremely accurate and precise. But, is that really necessary? Especially when the information of the figures is not properly described and discussed later…

Response:

Thank you for your comments. To report this relatively new finding and draw the conclusion with a rigorous attitude, we performed at least three replications of each experiment with independent biological samples. As a result, we presented precise quantitative data and SD values in the Figures. We apologize for not fully describing these results in the text. We have added a detailed description of the results (in the Results section) and made further discussion (in the Discussion section) in the revised manuscript.

3)  Please, consider the use of the same structure for abbreviations. Why Nar and not NAR? The authors use PSC, MYH, etc… but not Psc, Myh…

Response:

Thanks for your kind suggestion. We had changed “Nar” to “NAR” in the revised manuscript.

4) Lines 76-78. This paragraph does not belong to the Introduction. In this paragraph, the objective or the aim of the study must be described.

Response:

Thanks for your comments. According to your kind suggestion, we have rewritten the last paragraph of the Introduction section in the revised manuscript. We have also highlighted the aim of the study in this paragraph. The revised paragraph is as follows: “This study aims to find an economical and effective additive that supports the generation of a large number of highly mature myotubes in vitro. Furthermore, the relevant molecular mechanism and application strategy will be explored. These results contribute to the development of efficient myotube production systems to promote the pro-duction of high-quality cultured meat products with better public acceptance. It also provides a theoretical and technical basis for deepening muscle development.” (Lines 76-81)

5) Line 24. Porcine myotubes? How do the authors know that the myotubes are “porcine”?

Response:

Thank you for your kind suggestion. We are Sorry for the less rigorous description. We have rewritten the sentence in the revised manuscript as “Taking together, these findings suggested NAR induced the differentiation of PSCs and generation of mature myotubes through upregulation of the IGF-1 signaling, contributing to the development of efficient and innovative cultured meat production systems.”. (Lines 23-25)

6) Lines 110, 117, 120, etc… Please, add a space between the number and the units.

Response:

Thank you for your kind suggestion. We have corrected these errors in the revised manuscript. We are very sorry for these less rigorous descriptions.

7) Lines 89-176. The authors use the phrase “room temperature” too lightly

Response:

Thank you for your kind suggestion. We have changed "room temperature" to “25℃” in the revised manuscript.  

8) Lines 184-198. Further description of Figure 1 results and critical discussion must be included. Or have the authors published the comparison among flavonoids elsewhere?

Response:

Thank you for your comments. We have added detailed description about Figure 1 results as “As shown in Figure 1a, b, the addition of NAR resulted in the greatest increase in the proportion of MYH3+ cells, which was 3.18-fold (SD = 0.36, P<0.01) higher than the control group, followed by LUT (2.65-fold, SD = 0.77, P< 0.05), QUE (2.14-fold, SD = 0.28, P<0.05), GEN (1. 99-fold, SD = 0.39, P<0.05), and EC (1.73-fold, SD = 0.17, P<0.05). Moreover, the fusion index also increased in a similar trend to MYH3 expression, with the NAR treatment achieving the most 5.70-fold (SD = 1.20, P<0.01) increase (Figure 1a, c).” (Lines 188-194). In addition, we have supplemented relevant discussions in the revised manuscript as “Lee et al. found 20 μM of EC enhanced myogenic differentiation of C2C12 cells through stimulation of promyogenic signaling pathways, p38MAPK and Akt [29]. Gan et al. proved that GEN treatment promoted myoblast differentiation at a dose of 10 µM [30]. In addition, QUE, LUT, and NAR were also reported to be effective in promoting differentiation of different myoblast cells [22,23,31]. In this work, we first verified and compared the effect of five flavonoids on inducing myogenesis in vitro and found that NAR showed the most significant effect.” (Lines 392-398).

The comparison among flavonoids has also been reported in several published studies. In a study reported by our research team, four representative flavonoids including luteolin, chrysin, apigenin, and genistein were investigated for their effects on myoblasts in the aspects of proliferation, migration, and differentiation (Guan, X.; Pan, Z.; Xu, Z.; Zhang, S.; Tang, H.; Du, G.; Zhou, J. Natural flavonoid luteolin promotes the differentiation of porcine myoblasts through activation of PI3K/Akt/mTOR signaling. Food Bioscience 2022, 47, 101766). In another study reported by Guo et al., three flavonoids including quercetin, icariin, and 3,2′-dihydroxyflavone were chosen with multi concentrations to evaluate the effect on proliferation and differentiation of muscle stem cells (Guo, Y.; Ding, S.-J.; Ding, X.; Liu, Z.; Wang, J.-L.; Chen, Y.; Liu, P.-P.; Li, H.-X.; Zhou, G.-H.; Tang, C.-B. Effects of selected flavonoids on cell proliferation and differentiation of porcine muscle stem cells for cultured meat production. Food Research International 2022, 160, 111459).

9) Lines 209-220. Further description of Figure 2 results and critical discussion must be included.

Response:

Thanks for your suggestions. We have revised and added further description of Figure 2 results as “When PSCs were exposed to various concentrations of NAR for 12 h, the proportion of Ki67+ cells in each NAR treatment and control group was around 20%, showing no significant difference (Figure 2a, b). Besides, PSCs were cultured in PM supplemented with 0 to 100 μM of NAR for 3 days and the absolute cell number was recorded. The results showed that after 3 days of culture, there was no significant difference in the absolute number of all groups, which was about 1.1×106 (Figure 2c). Meanwhile, the cell viability was measured by the MTT assay. The MTT absorbance was similar for all groups (Figure 2d), indicating that NAR was not cytotoxic to PSCs up to 100 μM. Taking together, our data suggested that NAR did not show a proliferative effect on PSCs in vitro, but it was not cytotoxic to PSCs.” (Lines 210-219). In addition, we have supplemented relevant discussions about the effect of proliferation in the revised manuscript as “NAR is a flavanone compound widely existing in tomato, grapefruit, and Citrus fruits and is an approved food additive [39,40]. Our experimental results also proved that NAR was not cytotoxic to PSCs at up to 100 μM, implying the safety of its application in cultured meat production.” (Lines 398-401).

10) Lines 222-227. Treatment “0” is missing.

Response:

Thank you for your kind suggestion. We have added “0” in lines 222-227 in the revised manuscript. We are very sorry for the errors.

11) Lines 229-323. Further description of Figures 3-5 results and critical discussion must be included.

Thanks for your suggestions. We have revised and added further description of Figure 3-5 results as follows:

The revised and added description of Figure 3: “MYH3 immunofluorescence staining showed that the myogenic differentiation ability was weakened with the increase of cell expansion and passaging, but treatment with NAR at 10, 20, and 50 μM all up-regulated the differentiation efficiency of PSCs at each passage significantly. Among the three concentrations, 20 μM NAR showed the optimum effect, with the myogenic fusion index of P4, P6, and P8 PSCs reaching 183% (SD = 7%, P<0.001), 207% (SD = 10%, P<0.001), and 278% (SD = 21%, P<0.001) of the control group, respectively (Figure 3a, b). Furthermore, we detected the expression of two myosin heavy chain (MYH) isoforms, MYH3 (generated at the embryonic stage) and MYH8 (generated at the neonatal stage). The results showed that the addition of 20 μM NAR during the differentiation of P4, P6, and P8 PSCs increased the expression of both MYH3 and MYH8 significantly and consistently. The expression levels of MYH3 and MYH8 proteins in the 20 μM NAR group were more than 150% of that in the control group (Figure 3c-e).” (Lines 229-241). In addition, we have supplemented relevant discussions about the effect of differentiation in the revised manuscript as “Moreover, in the two-dimensional (2D) culture, we found 20 μM NAR promoted the myogenesis of PSCs at different passages and significantly increased the expression of MYH8, indicating the generation of mature myotubes.” (Lines 401-404).

The revised and added description of Figure 4: “In particular, PSCs treated with 20 μM NAR showed significantly elevated expression of ERβ in whole cells and the nucleus, which was 178% (SD = 34%, P<0.05) and 173% (SD = 15%, P<0.001) of that without NAR treatment, respectively (Figure 4b, d). It suggested that NAR enhanced the biosynthesis and activity of ERβ during myogenic differentiation.” (Lines 260-264). “Consistent with the expression of MYH and ERβ proteins, 20 μM of NAR showed the optimum up-regulating effect on endogenous IGF-1 and its down-stream signaling pathway. Treatment of PSCs with 20 μM NAR increased the expression of endogenous IGF-1 to 175% (SD = 18%, P<0.01) of the control group (Figure 4e, g), and upregulated the phosphorylation levels of IGF-1R, AKT, and mTOR proteins to 229% (SD = 69%, P<0.05), 223% (SD = 67%, P<0.05), and 236% (SD = 32%, P<0.01) of the control group, respectively (Figure 4f, h-j).” (Lines 268-274). In addition, we have supplemented and modified relevant discussions in the revised manuscript as “ERs are widely expressed in skeletal muscle and myoblasts [41]. In general, ERs and estrogen form dimers and then bind to estrogen receptor binding elements to stimulate target gene transcription and regulate cell proliferation or differentiation [42,43]. ERs are mainly composed of two isoforms, estrogen receptor α (ERα) and estrogen receptor β (ERβ), both of which have been reported to be involved in muscle atrophy [44], muscle regeneration, and hypertrophy [45-47]. In this work, it was found that the pro-myogenesis role of NAR was related to ERβ. ERβ Signaling is an important factor in regulating skeletal muscle and muscle satellite cells. Seko and colleagues indicated ERβ was indispensable for muscle growth and regeneration in female mice [48]. Ogawa et al. found daidzein down-regulated the expression of ubiquitin-specific pro-tease 19 through ERβ and increased skeletal muscle mass in young female mice [49]. Velders et al. proved that ERβ regulated anabolic pathway, muscle satellite cell proliferation, and immune response to affect the growth and regeneration of skeletal muscle in mice [45].” (Lines 405-417))

The revised and added description of Figure 5: “The addition of PHTPP significantly reduced the levels of ERβ in the nuclei of Ctrl and NAR-treated PSCs by 33% (SD =15%, P<0.01) and 61% (SD = 28%, P<0.01) of the original levels, respectively (Figure 5a, b).” (Lines 296-298). “The expression of endogenous IGF-1 in PSCs treated with the combination of PHTPP and NAR was 67% (SD = 7%, P<0.01) of that in PSCs treated with NAR alone (Figure 5c, e). Similarly, the phosphorylation levels of IGF-1R, AKT, and mTOR proteins all de-creased obviously to less than 70% of the original level upon the addition of PHTPP (Figure 5d, f-h).” (Lines 300-304). “we observed that compared with PSCs treated with NAR alone, combined exposure of PHTPP and NAR markedly inhibited myotube formation during PSC differentiation and reduced the fusion index from 57% (SD =3%) to 43% (SD =1%) (Figure 5i, j),” (Lines 306-309). In addition, we have modified relevant discussions in the revised manuscript as “It has been reported that agonists targeting ERβ possessed anabolic potency and stimulated the expression of endogenous IGF-1 [45]. IGF-1 is considered to be a factor closely related to skeletal muscle growth and hypertrophy. It can regulate protein syn-thesis and degradation, affect mitochondrial autophagy, and activate calcium signaling mediated by calcineurin [24,33,50]. By antagonizing ERβ with PHTPP, our study showed that during the differentiation of PSCs in vitro, NAR mediated nuclear transfer of ERβ signaling to upregulate the expression of IGF-1, promote the activation of IGF-1/Akt/mTOR anabolic pathway, thereby regulating myogenic differentiation and boosting myotube maturation.” (Lines 418-426).

12) Figure 3. D) and e) SD bar is missing in Control. Same Figures 4 and 5.

Response:

Thanks for your comment. We used the normalization method in the analysis of Western blotting data, exhibiting the fold change over the control group. Therefore, each time the value of the control group was defined as 1 with no SD. The specific data analysis steps are as follows: 1) The signals of target and housekeeping proteins in all lanes in the blots were quantified; 2) the target protein was normalized to housekeeping protein (α-Tubulin, GAPDH, or Histone H3) in the same lane; 3) the value of target protein in the control group was defined as 1 and the other group was normalized to the control to obtain the value of fold change. Then, the results of each target protein from three individual experiments were calculated to obtain the mean value and SD bar. A similar way of data processing and graphic presentation was also shown in the following literatures:

  • Ding, S.; Swennen, G.N.M.; Messmer, T.; Gagliardi, M.; Molin, D.G.M.; Li, C.; Zhou, G.; Post, M.J. Maintaining bovine satellite cells stemness through p38 pathway. Scientific Reports 2018, 8, 10808.
  • Riis, S.; Murray, J.B.; O'Connor, R. IGF-1 Signalling Regulates Mitochondria Dynamics and Turnover through a Conserved GSK-3β-Nrf2-BNIP3 Pathway. Cells 2020, 9, 147.
  • Guan, X.; Yan, Q.; Wang, D.; Du, G.; Zhou, J. IGF-1 Signaling Regulates Mitochondrial Remodeling during Myogenic Differentiation. Nutrients 2022, 14, 1249.
  • Guo, Y.; Ding, S.-J.; Ding, X.; Liu, Z.; Wang, J.-L.; Chen, Y.; Liu, P.-P.; Li, H.-X.; Zhou, G.-H.; Tang, C.-B. Effects of selected flavonoids on cell proliferation and differentiation of porcine muscle stem cells for cultured meat production. Food Research International 2022, 160, 111459.

We have added description about the data processing method in the Method section 2.7 “Protein expression was normalized to α-Tubulin or GAPDH or Histone H3, then normalized to the control samples and presented as a relative fold change with the control sample set to a value of 1. In addition, we also added the above explanation in the corresponding figure legends, such as “The bar chart presents the quantification of blot band intensity, normalized to GAPDH levels, and presented as a fold change relative to the Ctrl sample of P4, which was set to 1.”

13) Cite format of the Reference list must be reviewed. It does not follow the journal requirements.

Response:

Thank you for your comments. We have carefully examined the format of the references and completed the revision.

  • From: From: Guo, Y.; Ding, S.-J.; Ding, X.; Liu, Z.; Wang, J.-L.; Chen, Y.; Liu, P.-P.; Li, H.-X.; Zhou, G.-H.; Tang, C.-B. Effects of selected flavonoids oncellproliferation and differentiation of porcine muscle stem cells for cultured meat production. Food Research International 2022, 160, 111459.

To: Guo, Y.; Ding, S.-J.; Ding, X.; Liu, Z.; Wang, J.-L.; Chen, Y.; Liu, P.-P.; Li, H.-X.; Zhou, G.-H.; Tang, C.-B. Effects of selected flavonoids on cell proliferation and differentiation of porcine muscle stem cells for cultured meat production. Food Research International 2022, 160, 1114592.

  • From: Guan, X.; Yan, Q.; Wang, D.; Du, G.; Zhou, J. IGF-1 Signaling Regulates Mitochondrial Remodeling during Myogenic Differentiation. Nutrients 2022, 14.

To: Guan, X.; Yan, Q.; Wang, D.; Du, G.; Zhou, J. IGF-1 Signaling Regulates Mitochondrial Remodeling during Myogenic Differentiation. Nutrients 2022, 14, 1249.

  • From: Yoshida, T.; Delafontaine, P. Mechanisms of IGF-1-Mediated Regulation of Skeletal Muscle Hypertrophy and Atrophy. Cells 2020, 9.

To: Yoshida, T.; Delafontaine, P. Mechanisms of IGF-1-Mediated Regulation of Skeletal Muscle Hypertrophy and Atrophy. Cells 2020, 9, 1970.

  • From: Nikawa, T.; Ulla, A.; Sakakibara, I. Polyphenols and Their Effects on Muscle Atrophy and Muscle Health. Molecules 2021, 26.

To: Nikawa, T.; Ulla, A.; Sakakibara, I. Polyphenols and Their Effects on Muscle Atrophy and Muscle Health. Molecules 2021, 26, 4487.

  • From: Kitajima, Y.; Ogawa, S.; Egusa, S.; Ono, Y. Soymilk Improves Muscle Weakness in Young Ovariectomized Female Mice. Nutrients 2017, 9.

To: Kitajima, Y.; Ogawa, S.; Egusa, S.; Ono, Y. Soymilk Improves Muscle Weakness in Young Ovariectomized Female Mice. Nutrients 2017, 9, 834.

Round 2

Reviewer 3 Report

The authors have considered all the reviewer´s comments and improved their manuscripts considerably.